# 'A Return, a Mirror, a Photograph': Return Journeys, Material Culture and Intergenerational Transmission in a Greek Cypriot Refugee Family

Christakis Peristianis

Department of Social and Political Sciences, University of Cyprus, Nicosia 1678, Cyprus; cperis01@ucy.ac.cy

**Abstract:** During times of war, displaced families carry various material items that later serve as means for preserving the memories of lost homes and maintaining a sense of identity. In divided Cyprus, the use of material objects by people displaced before and during the 1974 Turkish invasion has been influenced by the opening of checkpoints between the two sides in 2003. This paper explores how different generations in my family reacted to and interpreted the rediscovery of a lost material item—a handmade mirror piece—during the return journey. It discusses how my mother located the item, photographed it, and placed its photograph in the family's photographic archive. During the research project upon which this paper follows from, both items re-emerged through my mother's storytelling about her experience of return, transforming the project into a form of intergenerational transmission. The paper portrays how the storytelling about the mirror piece and its photograph was interpreted differently by me and my mother, influenced by the different politics of memory. The paper also showcases the resourcefulness of refugee families in maintaining the memory of their lost homes, which simultaneously reveals their views and hopes regarding the political future of the island.

**Keywords:** material objects; Greek Cypriot refugees; family history and memory; return journeys; intergenerational transmission

## 1. Introduction

During times of war and persecution, displaced families often carry a range of belongings that, during exile, come to serve as more than mere material possessions. Auslander and Zahra (2018) contend that these objects take on a significance beyond their material form, serving as means for safeguarding the memories of inaccessible places and homes, while also maintaining a sense of identity and continuity with the past. The study of these objects and how they are used and talked about in domestic settings can unveil their diachronic meaning for displaced families and their role as conduits for the intergenerational transmission of memory.

In divided Cyprus, the utilization of material objects in domestic settings has become increasingly complex due to the unique circumstances of the conflict. The 1974 Turkish invasion of the island resulted in the displacement of thousands of Greek and Turkish Cypriots, who left behind their homes and resettled in their designated areas—the Greek Cypriot Republic in the 'south' and the Turkish army-occupied 'north'. A militarized de facto border, known as the Green Line, separated the two regions along ethnic lines, dividing the 'south' from the 'north'. However, on 23 April 2003, the border between the 'north' and the 'south' opened, allowing individuals across the island to cross to the other side for the first time since 1974. Since this development, thousands of Greek Cypriot 'refugees' seized the opportunity to revisit their places of origin and other significant cultural sites in the north (Mesaritou 2023).[1] During these return journeys, many rediscovered lost family objects and heirlooms, often seeking to reclaim them. This led to bewilderment and moments of confusion among Turks and Turkish Cypriots settled in the north, who had

appropriated these items and come to regard them as their own (see Bryant 2010; Dikomitis 2012; Navaro-Yashin 2012).

The paper explores how Greek Cypriot refugee families react to and interpret the rediscovery of lost material items during return journeys. It focuses on how different generations respond to the reincorporation of these items into their family history and memory. What meanings do these objects hold for the individuals connected to them, and how are such meanings shaped by the politics of memory prevalent on the island? The paper delves into these questions by examining the history of an item linked to my maternal family's displacement: a handmade mirror piece that my grandfather received as a gift but which the family had to leave behind during their escape in 1974. Upon returning to their occupied village in 2003, my mother unexpectedly rediscovered the mirror piece in a neighboring house. Unable to retrieve it, she captured its photograph, which she then stored in the family's photographic archive. Approximately a decade later, both objects 're-emerged' through the research project that this paper follows, and during my mother's storytelling about her experience of return. As a result, the family history project evolved into a form of intergenerational transmission, wherein the influence of the island's politics of memory became particularly manifest.

The paper draws from a series of oral histories with my mother and autoethnographic research in my family home, presenting our conversations about the family's return journey in 2003. It focuses on the stark contrast between our interpretations of what occurred during the return journey: the rediscovery of the mirror piece and the act of photographing it. The paper's analysis connects the different generational understandings of what happened during my mother's return journey with larger themes in Cypriot history and society: a politics of memory dominant in the latter 20th century that focused on themes of dispossession and anticipation of recovery (Bryant 2008; Papadakis 2003, 2008). Lastly, it makes an argument for the recognition of the resourcefulness of Greek Cypriot refugee families in preserving the memory of their lost home, which simultaneously reveals their views and hopes regarding the political future of the island.

The paper begins by outlining its theoretical framework and how it conceptualizes material items. It then explores how scholarly literature has addressed the nexus between displacement, materiality, and family memory. Following this, I elaborate on the methods employed for data collection, with a particular emphasis on positionality and reflexivity in oral history research. In the main part, the paper presents the conversation between my mother and me regarding the mirror piece and analyzes our conflicting interpretations of what occurred during the return journey. It focuses on pivotal moments of the mirror piece's history: being given as a gift, its abandonment during the escape, its rediscovery during the return journey, and the photographic act. Throughout the main part, I reflect on our different meanings and reactions to the narrative event and how they are connected to the wider politics of memory on the island. The last section concludes the discussion, reflecting on the resourcefulness of refugee families in maintaining the memory of their lost home and how that is connected to their understanding of the island's future.

## 2. Considering Material Objects and Photographs

The paper draws upon the social biography approach (Appadurai 1986; Kopytoff 1986) and its recognition of material objects, not merely as outcomes of natural processes but as products of human activity.[2] Research following this approach argues for the possibility of tracing an object's biography and exploring its social life by examining the values and attitudes attached to it over time. The social efficacy of objects is grounded precisely in their changing roles and meanings as they are projected into different spaces to accomplish various tasks (Edwards 2012). This dynamism allows material objects to encompass dimensions that go beyond the intentions of their original design. As noted by Saunders (2004), objects may be ascribed significance that transcends their literal or functional purpose, shaped by specific personal, social, and historical circumstances and contexts. Thus, the paper delves into the social biography and cultural value of both the mirror piece and the

photograph captured by my mother by exploring the dynamic interplay between their production, possession, use value, and their role and meaning within the family setting and across generations.

### 3. (Forced) Mobility, Material Culture, and Family Memory

As Auslander and Zahra (2018) assert, "mobility and material culture have been inextricably intertwined throughout modern history" (p. 6), with each significantly influencing the other. Material objects can serve as symbols of a former home, one's past life, and loved ones left behind, playing a pivotal role in shaping one's evolving sense of identity. However, the introduction of a third element in the form of family memory and intergenerational transmission redefines the relationship between (forced) mobility and material culture, offering an opportunity to explore the dynamics of memory and identity that extend beyond the individual. Discussing the theme of home, Blunt and Dowling (2022) draw attention to homes as sites of memory where personal and inherited connections to a former home can be enacted and reworked. Consequently, family memory reshapes the relationship between (forced) mobility and material culture, bringing family dynamics and intergenerational transmission into greater prominence.

Much of the literature on mobility and material culture concentrates on the diasporic dimension of migration, particularly the role of material cultures in diasporic homemaking. Researchers such as Rose (2010) and Tolia-Kelly (2004) have delved into how family photographs evoke the memories of past homes, revealing broader connections between material culture, homemaking, memory, and identity. Other scholars have examined items such as paintings, DVDs, and plastic bowls (Walsh 2006). More recently, Sabine Marschall (2019) explored intra-African mobility, discussing the absence of sentimental mementoes and the use of 'utilitarian' objects, often given as gifts, that function as 'memory objects' for migrants.

Significant contributions to the theoretical and conceptual framework of the relationship between mobility and material culture have been made by authors such as Kathy Burrell, Maja Povrzanović Frykman, and Maruška Svašek. In her contribution to a special issue for the journal *Mobilities*, Burrell (2008) examined the materialization of journey and border times through items like passports, cars, coaches, suitcases, and laptops in airport lounges, highlighting the intersection between movement and materiality. Moreover, Maja Povrzanović Frykman (2016a, 2016b) has suggested that migration research should closely examine the practices and lived experiences involving objects carried, sent, received, and used by (forced) migrants. She outlines an ethnographic, material-centered approach with three theoretical dimensions: the presence of objects in specific locations, the continuity of practices perceived across borders, and the practice-based sense of emplacement. Additionally, Maruška Svašek (2010, 2012), focusing on the emotional dynamics of migration, presents a framework that captures the different object-subject relationships that develop during mobility, employing the terms 'transit', 'transition', and 'transformation' in her analysis. These terms relate to the movement of people and things through time and space; the changes in the meaning, value, and emotional efficacy of objects and images related to transit; and the changes in individuals relate to transit.

Important inputs addressing materiality and forced mobility also include Sandra H. Dudley's (2010) monograph *Materializing Exile* and Dossa and Golubovic's (2019) exploration of the connections between narratives and textures of home. Dudley's book examined the effects and meanings of objects among Karenni refugees, exploring how these refugees perceive, represent, and engage with material objects and spaces. Conversely, Dossa and Golubovic (2019) investigated the relation between narratives of home that "nurture a sense of home through memory and storytelling" and textures of home that sustain it "through material objects, sometimes deeply personal. . . sometimes inconsequential" (p. 172). They offer an example of how 'inconsequential' material objects may sustain a texture of home by presenting the case of Palestinian refugees who pass down their house keys through generations (Dossa and Golubovic 2019). As they note, the house keys remain

the only tangible link to the lost home, and their passage within families serves as a tangible representation of ownership.

Research into the role of material objects in family memory has extensively examined the use of photographs and family albums in domestic settings. From Chalfen's (1987) analysis of the diverse uses of personal photography in American homes, to Freund and Thomson's (2011) investigation into the use of photographs and their connection to (oral) history, as well as Kuhn's (2002) analysis of pictures from both her own childhood and a more collective past, various works have delved into how family memory operates through material objects, particularly photographs. Perhaps the most notable example of works connecting material culture and family memory has been Hirsch's (2012) work with 'postmemory'. Hirsch introduced the concept to capture the relationship that the descendants of Holocaust survivors have with the personal, collective, and cultural history of their predecessors and the way it significantly impacts their identity. She argues that the descendants of the survivors of traumatic events establish deep connections with their parents' past through the stories, images, and behaviors that they have grown up with. Simultaneously, Hirsch (2012) acknowledges the significance of the family as an institution in this process, recognizing it as "an accessible lingua franca, easing identification and projection across distance and difference" (p. 115).

## 4. Methodology

This paper is part of a larger research project that examined the memories and meanings of home within my maternal extended family, who were internally displaced in 1974. Like many Greek Cypriot rural households, the family was large and relied on livestock and agriculture for their livelihood. Following displacement, most family members relocated to Nicosia, finding employment in diverse sectors. The study involved 28 participants spanning two generations: 14 'historical eyewitnesses' who had lived through the Turkish invasion and 14 individuals classified as the 'second generation'. Fieldwork took place in Cyprus from 2016 to 2017, organized into four two-week stages, each involving interviews with two nuclear families. The research questions focused on how parents conveyed their war experiences to their children, the impact of displacement on family dynamics, and how family members remembered and connected with the properties that they had left behind.

Diane L. Wolf (2020) observes that researchers' interests and their personal and family histories have a mutually constitutive relationship. This dynamic is also evident in my own interest in forced migration and family dynamics. During the 1974 Turkish invasion of Cyprus, my entire maternal extended family was displaced. This personal and familial experience significantly influenced my choice of research topic: exploring the memory of forced migration within the context of the extended family. However, it proved significantly challenging to overlook the parallel presence of the personal experience and of the social, political, and historical setting of what I wished to examine. When I discussed this topic with my supervisor, I was fortunate that he was also researching a similar subject at the time and was supportive of my idea. With his approval, the sociohistorical aspect became intertwined with the personal.

It is undoubtedly vital to recognize how the deep personal connection in this research project influenced the knowledge produced. Being a member of the family under study brings a distinct 'insider' perspective to the research. I agree with Akemi Kikumura (1986) that this personal connection provides special insight into the matters and feelings described, while often also bringing with it assumptions of prior knowledge and biases. Simultaneously, the analysis in the paper reveals my own heightened vulnerability in the research relationship and in the topic discussed. As Alan Wong (2013) maintains, different ways of listening are shaped by the varying relationships that we have with our narrators; this is evidenced through my 'intense' reactions to my mother's storytelling. Moreover, the narratives generated in this context are indicative of the way that contributors come to understand their seemingly individual experiences as interdependent and interconnected (Bochner and Ellis 2016; Pensoneau-Conway and Toyosaki 2011). As will be shown

throughout the paper, while the narrated event revolves around the experience of the return journey of my mother, the narrative event and my reactions to it co-constructed an entirely new sociocultural issue related to different generational interpretations and the politics of memory on the island (Borland 1991). As such, while the individual who experienced the events described may dominate the discussion, they are by no means alone in the construction of knowledge.

The nature of the research project raised several ethical issues. Initially, family members were more inclined to participate due to the family relationship. At the same time, confidentiality issues arose when the participants disclosed sensitive information, which had to be deliberately omitted to prevent potential harm to family relationships. However, the most significant ethical challenge revolved around the informed aspect of consent. The participants would sign consent forms and provide verbal consent without prior knowledge of the study's details. An illustrative example was when my mother signed the informed consent form without reading it and then chuckled when I attempted to explain its significance.

The project employed a variety of research methods, including oral/life history interviews, home observations, and a return journey to the family's ancestral village, the first and only time I have visited my maternal family's home in the north. This paper primarily relies on two oral history interviews conducted with my mother in our family home in Nicosia early in the fieldwork and our discussions about her return journey in 2003. It integrates family narratives by both me and my mother with the analysis of a photographic image, making use of the latter as both evidence and as a tool that elicited storytelling (Freund and Thomson 2011). The interviews were recorded, transcribed, and then analyzed using an approach that viewed material items as generating symbolic attachment and connecting with identity across space and time (Saunders and Cornish 2009). Lastly, an integral part of the paper involves my continuous reflections on positionality, influence on my mother's narrative, and assumptions related to the experiences described, making reflexivity a crucial aspect of the analysis (Finlay 2002).

## 5. Return Journeys, Material Objects, and Family Memory

Members of my maternal extended family visited their ancestral village on two separate occasions in 2003. During the first week after the opening of the checkpoints, my eldest maternal aunt, Hera, and her husband visited the village, the family house, and the family's orchards. A week later, all six female siblings of the family, accompanied by some of their husbands and children, decided to cross together during the weekend. Unfortunately, I was unable to participate in the return journey due to my engagement in military service at the time. The first to arrive at the house on that day was my third eldest aunt, along with her husband and one of her sons. Her extreme reactions to the house's condition displeased the occupants to the extent that they did not allow the rest of the five siblings and their families to enter. As family members stood outside their former house, unable to gain entry, an elderly Turkish Cypriot woman living across the street invited them into her own home. It was there that they discovered the mirror piece furniture.

Conveying the significance of the mirror piece for the family presents a formidable challenge. It was the sole material object connected to the family's history that most family members encountered during their return journey, a fact that undoubtedly added weight to their narratives surrounding its rediscovery. Furthermore, as discussed elsewhere (Peristianis 2023), the house that they came across during the return bore no resemblance to the home they remembered, challenging their sense of belonging there. The discovery of something familiar and meaningful, albeit in a different location, certainly heightened the importance of this rediscovery. Lastly, it is important to note that family members discussed the mirror piece and its rediscovery in varied ways. While some offered only a brief description of the object in their narratives before emphasizing its rediscovery, my mother stood out as the sole sibling who underscored its physical characteristics and its

connection to the family's history. This suggests, I believe, that she attributed additional significance to the object and its connection to the family's history compared to her siblings.

The interviews conducted with my mother were the first ones to occur for the entire research project. Much of the information that she shared about the return journey, the rediscovery of the mirror piece, and the aftermath was later complemented by details provided by other family members. Nevertheless, the paper's main focus is on the narrative event—the conversation between my mother and me—regarding the mirror piece and my reactions to her account and interpretation of events (Borland 1991). Details from the interview with my eldest aunt, Hera, provide additional information regarding the conversation that transpired with the elderly Turkish Cypriot woman following the rediscovery of the mirror piece.[3] The section is divided into four parts, each focusing on pivotal moments of the mirror piece's history: being given as a gift, its abandonment during the escape, its rediscovery during the return journey, and the photographic act.

### 5.1. The Gift

The first mention of the mirror piece during the interviews with my mother came when I inquired about her return journey. Following questions about the decision to cross for the first time and the people whom she went with, I asked about whether they had found any of their old belongings upon their return to their former house in the north. Although the initial response was "no", she quickly transitioned to describe the mirror piece. This moment in the narration became confusing as she transitioned from asserting that they had found nothing in their former house to describing the mirror piece:

> "No! Ehm, we had a mirror with 'kalimera' [good morning] written on it, painted. Somebody had made it for your grandfather, and it was a very special thing. A mirror that had 'kalimera' written on it, with painted things."

The piece had a rectangular shape and featured a heart-shaped mirror at its center. It included calligraphic writing of the word 'kalimera' on top of the mirror, while painted flowers on a pink surface on each side of the mirror added a decorative element to it. My aunt, Hera, also mentioned that the piece served as a coat rack for hanging their coats and was placed in the living room, right by the entrance of their house.

During her narration, my mother emphasized that the piece was a gift for their father, specifically crafted for him by someone in the village. I interpreted this emphasis as an attempt to highlight the high regard that their father enjoyed in the village, a sentiment that she had also previously commented on during her interview. As Herzfeld (1980) describes, 'having respect paid' was a socially sanctioned ideal in rural Mediterranean societies and treating certain individuals as privileged signified positive public evaluation. Through her narration, my mother portrayed the mirror piece as evidence of this positive public evaluation and her father's "adherence to and enactment of the moral code of expectations" in the village (Anagnostou 2021, p. 8). This object was thus symbolically linked to her father's identity and social position, with this significance encapsulated by her description of the piece as a "very special thing".

### 5.2. The Flight

Earlier in the interview, my mother narrated how the family left the village on 14 August 1974. She recalled attending the morning church service, being the day before the religious holiday of the Assumption of Mother Mary. In church, discussions about staying or leaving the village were prevalent. However, her father was adamant about staying, firmly believing that "the Turks would not come to the village". Nevertheless, in the late afternoon, the entire family boarded the elder sister's car and headed south towards the Troodos mountains. They left completely empty-handed, prompted by a neighbor's reprimand directed at their father for not leaving earlier, especially given that he had five unmarried daughters.[4] One of my mother's remarks underscores the hastiness of their departure: "when the afternoon came, we all entered my elder sister's car, and just left. We thought we would return in two or three days".

The belief in the temporary nature of the escape was a defining aspect of the Greek Cypriot experience of displacement (Loizos 1981). This belief profoundly influenced how refugees left their homes and what, if anything, they brought with them. I have heard stories of people leaving their homes with coffee cups and newspapers still on the kitchen table, and trousers left on the chair, as if they were stepping out for a walk. Therefore, it is crucial to note that the abandonment of possessions by Greek Cypriots in 1974 was not due to the material qualities of objects nor the absence of personal and familial significance. Instead, it was precisely the perception of the escape as being temporary—a routine act of 'going out of the house' with the expectation of a swift return—that led to empty-handed escapes and belongings left behind.

*5.3. The Return*

It was during the account of the return journey and the rediscovery of the mirror piece in the house of the elderly Turkish Cypriot woman that my mother's and my interpretations of the events described began to diverge. This is evident in how I kept questioning the actions described, as they did not align with my expectations of how Greek Cypriots should act regarding their lost properties.

> SV: We entered the house, and we did not see anything of ours. ... But we entered the house opposite, when she was calling us "come, come here"; and we entered, and we saw the mirror with the 'kalimera' in the house of the 'yiayia' [grandmother] across. Of the 'yiayia' across our house. The Turkish Cypriot 'yiayia' that stayed there.
>
> CP: Oh, did she take the mirror from yours?
>
> SV: She had taken it from the house and put it on top... we went to the toilet, and it was inside. And we told her: "this is ours".
>
> CP: And you did not want to get it?
>
> SV: No. We did not want to get it.

The rediscovery of the mirror piece, as documented in the above exchange, was understood and interpreted differently by my mother and me. On one hand, my mother seemed to focus her narration on how the family managed to reconnect with an item from their history, despite its spatial displacement. She emphasized that the Turkish Cypriot woman's appropriation of the object inadvertently aided her family in locating something of theirs during the return journey. The way that she animatedly described this act and her use of the term "yiayia", which signifies vulnerability and respect in colloquial Greek, to describe the Turkish Cypriot woman, suggested that she was attempting to justify her appropriation of the item. It was thus a memory focused on the family, the object, and the reconnection between the two.

On the other hand, my interpretation of the narrated events was centered on the Turkish Cypriot woman's appropriation of my family's possessions. To me, it seemed like someone from a neighboring house had entered my family's property in their absence, taken an item, and placed it in their own house. This action was interpreted as theft or looting, prompting me to seek my mother's confirmation of this interpretation, querying whether "she had indeed taken the mirror from our house". It was thus a postmemory focused on dispossession and appropriation.

Harald Welzer et al. (2002) state that family memory is continuously fine-tuned, but all family members' memories are coherent. The only exception to this fine-tuning is when family memory is threatened by a 'wrong' memory. This was the case with my own interpretation of the narrated events, which was influenced by a politics of memory that had emerged on the Greek Cypriot side of the island following the invasion. As various authors have documented, the collective memory that arose in the Greek Cypriot south institutionalized '1974' as a wound, manifested in the partition of the island and the politicization of personal suffering (Bryant 2012; Bryant and Papadakis 2012; Papadakis

2003, 2008; Roudometof and Christou 2014). History itself was engaged as an actor in the conflict, from history education and school textbooks (Bryant and Papadakis 2012; Papadakis 2008) to the historical narratives of the struggle in museums in the divided city of Nicosia (Papadakis 1994). This socio-cultural understanding of '1974' overshadowed all areas of social life on the island, with the representation of victimhood being evocative of an absence that anticipated a healing "in a future where all wrongs are set right" (Bryant and Papadakis 2012, p. 8). Regaining 'our' lands and properties in the north was thus a central component of these politics of memory, at some point even becoming more prevalent than the notion of collective suffering (Rakopoulos 2022).

These were the politics of memory that I encountered during my formative years. From images of villages in the north with the slogan "I do not forget" on school textbooks and walls, to mottoes such as "under the right circumstances, they will be ours once again", these messages and representations were prevalent in all areas of social life, from school classrooms to media and football stadiums. This pervasive influence of the politics of memory on the island became evident through the disparity between my mother's and my interpretations of what occurred during the return journey. Nevertheless, it was not only the appropriation of the mirror piece by the Turkish Cypriot woman that was interpreted differently, but also the family's claim over the object. My impulsive questioning regarding whether the family did not want to retrieve it was met with a remorseful negative reply. It seemed that my mother was apologizing for not conforming to what Greek Cypriot society and the politics of memory (through me) 'expected' her to do: reclaim her property.

As mentioned earlier, my mother was the first person whom I interviewed for the research project, and her interview marked the first time that I had heard about what transpired during the return journey. My spontaneous reactions to her storytelling shaped a narrative event (Borland 1991) where our generational differences and, more specifically, the influence of the politics of memory on the island became particularly manifest. For my mother, the story of the mirror piece and its rediscovery during the return journey were narrated through a familial lens of remembrance (Welzer 2008). It was the story of a family item, which was rediscovered during their return journey in a neighboring house. For me, however, this story was interpreted through a politics of memory that emphasized dispossession and the expectation of recovery. It was the story about a material object that had been stolen by the 'other', and upon its rediscovery, the family had failed to reclaim.

Nevertheless, the most striking aspect of my mother's account was her assertion that they did not wish to reclaim the object. Indeed, I remember that her narrative left me with the impression that the family had requested the return of the object, and the Turkish Cypriot woman had declined, with her response serving as a way to mask this failure. During the second phase of the fieldwork and the interview with my oldest aunt, Hera, this assumption was confirmed. Hera explained:

And I told her, I told her at that moment: "this item is ours", I told her! I knew that that house did not have that item. And I told her: "this item is from the house across, our house, that they brought it". She did not give it to me. "I don't know" she said. We told her but... "I, when I came," she said, "it was here".

Similar responses by Turkish Cypriots to Greek Cypriots claiming back their rediscovered objects during their return journeys were reported by Rebecca Bryant (2014). In similar scenarios, she elaborates on the account of a Turkish Cypriot woman who refused to return a clock to the Greek Cypriot owner of the house that she was occupying, claiming that "it's the house's clock," as well as the story of a Greek Cypriot who was denied the return of a painting that her father had bought (Bryant 2014, p. 32). Bryant (2014) refers to these objects as 'remainders' of history, enigmatic objects that are no longer clearly linked to their original owners and have been appropriated in the post-1974 period. The mirror piece had become such a 'remainder', as it was made to fit into new settings. As seen in Figure 1 below, the Turkish Cypriot woman in whose house the object was located had placed it on a wall next to her own family photographs. The mirror piece had found a new sense

of appropriateness, embedded in new family contexts and acquiring new sets of social expectations and desires.

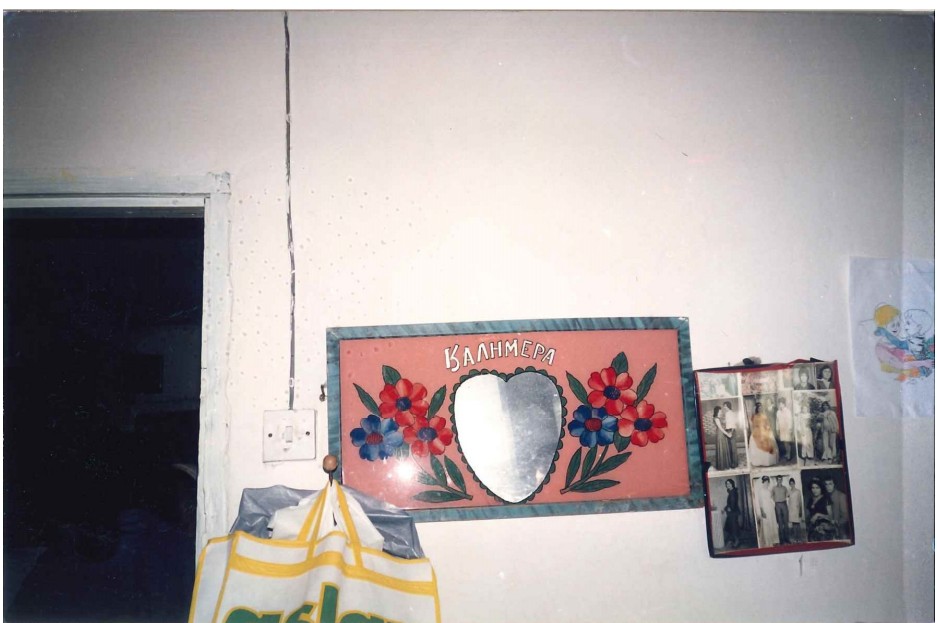

**Figure 1.** The photograph my mother took upon rediscovering the mirror piece during the return journey. The object is seen situated alongside vintage family photographs of the Turkish Cypriot woman and other drawings, with a hanging grocery bag below it.

*5.4. The Photograph*

Listening to my mother's narrative about the appropriation of a family item, its rediscovery during the return journey, and the 'unwillingness' of the family to reclaim it were quite surprising. However, what proved even more surprising was that she continued her narration, as she casually commented that she "took its picture and still had it to that day".

People tend to take photographs to capture important events and experiences that they want to remember and cherish. I would argue that the photograph that my mother took was precisely meant to capture that moment of rediscovery, the reconnection with an object linked to the family's history, which they had no access to before. This reconnection went beyond mere recognition of something familiar; it symbolized a link with the family's life prior to 1974. As Tolia-Kelly (2004) reminds us, "solid materials are charged with memories that activate connections" (p. 314) to past lives and family members. The urge to photograph the mirror piece upon its rediscovery was thus an affirmation of its enduring 'stickiness' (Ahmed 2010), that affective quality enabling objects to draw on familiar connections to memory and identity. To put it plainly, the photograph served as evidence that the mirror piece could evoke emotional connections to a life before 1974.

Since 2003, the photograph has assumed multiple roles: of an 'index', an 'icon', and a 'symbol' (Erll 2011). Firstly, it now ascribes truth-value to the moment of rediscovery, confirming its actual occurrence. Secondly, it represents the visual essence of the past. This is evident in the way that I could reproduce the visual features of the mirror piece and its surroundings—something that I could not achieve solely through the narratives of my family members. However, it is what it stands as a symbol of that proves most intriguing. The photograph simultaneously represents the resourcefulness of refugee families in preserving the memories of their lost homes as well as their views regarding the political future of the island. On one hand, the photograph serves as evidence of refugees' creative and practical methods in keeping the memory of their lost homes. My mother's photograph offers a lasting and easily transportable representation of home, ensuring

that its memory and legacy persist and are passed down to future generations. Indeed, following her mention of the photograph, I inquired why she had never shown 'it' to me, to which she responded that she has kept 'it' in the photographic archive from the return journey. During that brief exchange, we discussed 'it' as if referring to the mirror piece itself, rather than the photograph, as evident in the original Greek transcription.[5] Thus, the photograph became a representation of the mirror piece and the home left behind.

On the other hand, the photographic act illustrates also the enduring struggle that refugees have faced in coping with the longstanding and unresolved dispute on the island. Efforts to find a solution have been ongoing for decades, involving numerous negotiation attempts, international interventions, and peace talks (see Varnava and Faustmann 2008). However, each period of optimism and progress has ultimately led to a stalemate. This unpredictability regarding the resolution of the dispute has left refugees in a state of uncertainty for over 50 years, torn between the memory of a life prior to 1974 and the necessity to adapt to new circumstances. I believe that the photograph that my mother took during the return journey represents also a realization of the obstacles to witnessing a resolution to the dispute, at least during her own lifetime (she is now in her late 60s). It is essentially a symbol of the uncertainty characterizing the Cyprus dispute, where refugees have taken to photographing their properties and displaying these photographs in their new homes, not knowing whether or when they will be allowed to return to their places of origin.

## 6. Conclusions

Family history research is primarily motivated by three factors: validating descendancy for membership purposes; immersing the researcher into a relatively obscure past with the intention of investigating it; and expressing a past as a means of gaining individual and transpersonal identity (Barnwell 2013). My research into my maternal family's history of displacement was motivated primarily by the latter two. On one hand, while the extended family was of significant importance in the regions from which Greek Cypriot refugees originated, previous research had not sufficiently delved into how displacement had impacted this institution (see, however, Loizos 1981). Acknowledging the latter's importance for Greek Cypriot refugees was a way of recognizing a significant form of cultural heritage and examining how it was affected by displacement. On the other hand, I saw a study into my own family's past as a way of expressing gratitude for everything that the older generations had done for my contemporaries, despite the hardships and losses that they suffered. Additionally, it was a form of recognition and honor for my maternal grandparents who had died in displacement, without ever returning.

Harald Welzer (2008, 2010) reminds us that history extends beyond books, curriculums, and public debates; it is also woven into the fabric of everyday life and the way that experiences and memories are passed down within family contexts. In this instance, history revolves around my maternal family's ordeal of displacement and their return journey to their place of origin, approximately 30 years after their escape. History is the tale of a material item left behind, rediscovered, and memorialized during the return journey. History is also about the narratives and dialogues within the home about that material item. However, conveying history is not merely a process of transmitting fixed contexts; instead, history is an intersubjective process (Welzer 2008). While the individual who experienced the events described may dominate the discussion, they are by no means alone in the construction of knowledge.

This paper explored the story of a material item linked to my maternal family's history of displacement. It delved into the journey of a mirror piece furniture, which was gifted to my grandfather prior to 1974 but left behind during the family's escape in 1974. The same object was rediscovered in another house during the family's return journey in 2003. Subsequently, my mother photographed the object and placed the photograph in her photographic archive. The paper first investigated how the actions connected with the mirror piece were interpreted differently by me and my mother during her storytelling.

The contrasting perspectives and reactions to the storytelling highlight the influence of a politics of memory prevalent on the island following 1974, which impacted younger generations and their interpretations of the island's history. While my mother saw the story as the rediscovery of a family item during the return journey, my interpretation was shaped by a politics of memory emphasizing loss and the anticipation of recovery. It focused on what I perceived as a stolen family material object discovered in another house during the return. As such, the politics of memory have influenced the intergenerational interpretation of the island's history, whether it be the national history presented in books and public debates or the familial history passed down within families.

My mother's response to the rediscovery serves as evidence of the resourcefulness of refugee families in preserving the memory of their lost homes. When she found the mirror piece, she took a photograph of it, essentially generating a new (memorial) object in the form of a picture. This picture now serves as a commemoration of her previous home and provides an immediate link to the past while she resides in her new home in the south. The picture represents her quest for continuity and meaning associated with the past home, while also enabling her to progress with her current life. Simultaneously, the photographic act also marks a realization of the obstacles to witnessing a resolution to the dispute, at least during her own lifetime. In essence, the photograph embodies the negotiation between continuity and change which lies at the heart of the refugee experience.

**Funding:** This work was supported by the Arts and Humanities Research Council (AHRC).

**Institutional Review Board Statement:** The study was approved by the Ethics Committee of the University of Essex.

**Informed Consent Statement:** Informed consent was obtained from all subjects involved in the study.

**Data Availability Statement:** The raw data supporting the conclusions of this article will be made available by the author on request.

**Acknowledgments:** I would like to thank the three anonymous reviewers for their helpful comments on earlier drafts of this paper. Furthermore, I want to thank Michael Roper for his supervision of the research project this paper follows from.

**Conflicts of Interest:** The author declares no conflicts of interest.

## Notes

[1]    The vast majority of Greek Cypriots who fled their homes in 1974 settled in the south of the island and never crossed international borders. Nevertheless, the term 'refugees' has been widely used to describe them in Cypriot discourse, and it will be the term used here as well (see Zetter 1991; Roudometof and Christou 2014).

[2]    Studies following this approach have examined material cultures such as guns (Pearson and Connah 2013), 'trench art' (Saunders 2003; Saunders 2004), or items produced during captivity (Mytum and Carr 2012; Dusselier 2008).

[3]    Any names presented in the paper are pseudonyms.

[4]    My mother later connected this point with widespread reports of the rape of women and girls by advancing Turkish troops (see Loizos 1981).

[5]    In Greek, a mirror is a male noun (*ο καθρέφτης*, "o kathreftis"), while a photograph is referred to in the female gender (*η φωτογραφία*, "i fotografia"). The use of 'it' in the brief exchange corresponded to the male pronoun.

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
