# Peer review of "‘A Return, a Mirror, a Photograph’: Return Journeys, Material Culture and Intergenerational Transmission in a Greek Cypriot Refugee Family"

_genealogy, doi:10.3390/genealogy8020057_

Round 1
Reviewer 1 Report
Comments and Suggestions for Authors
Based on autoethnographic research with the maternal extended family, the author explores the significance of lost objects after displacement, questioning their different meanings for different family members ('first and second generation'). The author does so by zooming in on the rediscovery of a mirror upon a return visit by his mother, the picture taken of this mirror and kept in a family picture archive, and the narrative conveyed about the meaning of the rediscovery of the mirror, in a conversation between mother and author. Although I have some minor remarks about the first sections of the paper (mainly textual, see below), my suggestions primarily concern the sections that engage with the research findings and analysis.
In these sections, significant attention is given to the various meanings the mirror and its picture represent as 'remainders' and 'reminders' of the past (section 5). This is well explained and convincing. Yet the next section seems more fascinating/innovative. Section 6, now titled 'family history, cultures of remembrance and intergenerational transmission' also raises more questions/ is less well unpacked:
In section 6 the author directs us to a fascinating argument that I would like to see receive more attention and unpacking in this manuscript. Namely, how the influence of societal cultures of remembrance, or rather, I would maybe say a 'politics of memory', informs different ways of understanding the past (for different generations), mediated through lost/re-discovered objects. The point is important also to better connect with theory about intergenerational transmission in the aftermath of disaster/war/violence (to which the author refers fleetingly, but without going in-depth.) The findings could help understand the mechanisms that shape continuity and change over the generations - and see why, through which mechanisms, the knowledge changes in the discussion/transmission process: page 10 of the manuscript "While the individual who experienced the events described may dominate the discussion, they are by no means alone in the construction of knowledge."
To expand/unpack this argument better, the author could:
- contextualize by elaborating/ describing the societal (political) culture of remembrance/ politics of memory (see comment about this concept below) among the Greek Cypriots? This is now only referred to briefly, but seems a very important influence.
- Connect explicitly upon how this politics of memory may have informed the author's/second generation's interpretation of the past, especially meaning of lost/re-found objects?
I feel these elements are in the manuscript, but need more explicit attention/elaboration.
In addition, I have several comments about the text/ use of terms/concepts:
line 15: Missing words "the project into ... intergenerational transmission ... " (a form of?, experiment for? ,...... and transmission of? the past/ family history/ history?)
ln 16: words missing: oral history interviews
ln 20: familial and societal cultures of remembrance: I would rather propose to refer to the concepts 'politics of memory' and 'vernacular memory practices'? I am especially not convinced of the 'familial culture of remembrance' - e.g. the author's own practices of remembrance, although of the same family, differ from his mother's.
ln 71: oral history interviews
ln 79-81: see comment about familial culture/ societal culture. (in the conclusion, the author talks of the familial lens of remembrance, this already more seems more accurate)
ln 130: wording: "elements such as ..." (can one call family dynamics and intergenerational transmission 'elements'?)
ln 266 and later: 'refugee house': Is this a common term to refer to a house previously owned by people who lost the house when they became refugees? For me it is a confusing term, as the house is not (anymore) inhabited by the refugees, and when the house was theirs to live in, they were not refugees.
line 402: is this a direct quote? Confusing because of the noun "she"
ln 420: Why are the '...' used to talk about the current owners of the mirror piece?
521: ganimet (also mentioned earlier) Is this meaning for the Turkish Cypriot neighbour projected (speculated), or how does the author know that is the meaning attached? Could their also be a different meaning, given that the mother's meaning also differs from what appears more common (official) in Greek Cypriot society?
526: The author mentions (also in the beginning) that the mother's reaction to the rediscovery is "uncommon". This is puzzling in a way, and perhaps helps to answer the question about the generational juxtaposition sketched in the analysis --> Is it 'uncommon' compared to meanings attached by Greek Cypriot society at large, so likely influenced by the politics of memory/societal lens of remembrance? Or 'uncommon' for people in society but not in her family (familial lens of remembrance)? Or perhaps common for people of her generation, but uncommon for the second (and third,...) generation, demographically more significant? Or only seemingly 'uncommon' because these understandings are relegated to the vernacular/private spaces, away from an official politics of memory?
I believe the manuscript can potentially contribute more to the literature when these puzzles (section 6) receive more attention/ unpacking. Yet overall, the manuscript is well written and relevant for the special issue and journal.
Comments on the Quality of English Language
no comments
Author Response
Response to the reviewer:
Firstly, I would like to thank the reviewer for their comments. All are well-received and I believe the new version of the paper, which incorporates these points, is much stronger. See below:
- I have re-contextualised the entire discussion and analysis of the paper. I agree that too many points and discussions were running parallel in the paper. The new version of the paper focuses on a) the conflicting interpretations of the story/narration about the mirror piece between my mother and me and how these interpretations were influenced by the politics of memory on the island prevalent when I was growing up, and b) the way the photographic act is connected to refugees’ understanding of the island’s future. These two discussions have their own sub-sections (‘the return’ for a; ‘the photograph’ for b). It must also be noted that the sub-section ‘the return’ provides more context for the politics of memory and describes its main parameters.
- Similar to the point above, I have made the point about how my interpretation of the story was influenced by the politics of memory on the island clearer in the new version of the paper.
Comments about use of terms/concepts:
- Line 15: comments addressed – “… form of intergenerational transmission”
- Line 16: comment addressed – phrasing of the abstract has changed
- Concept use: I have used the concept politics of memory throughout the paper, replacing the term “societal culture of remembrance”
- Line 71: comment addressed
- Line 130: comment about “elements such as…” has been addressed – deleted term elements
- Term’s use: I have referred to the “refugee house” in a different way (either referring to it as “their former house” or “house in the north”) in order to avoid confusion
- Line 402: I have changed the phrasing of the quote in order to avoid confusion
- Line 420: Comment addressed - I have changed the term to holders
- Concept/term use: This point and the unclarity regarding the use of the term ‘ganimet’ was something all reviewers commented on. The new version does not make any reference to ‘ganimet’, precisely because it did not fit in the new argument and focus of the paper.
- Use of term “uncommon” throughout paper: The term “uncommon” was used with two different meanings in the original version of the paper.
- In the introduction, it referred to the general use of material objects in the context of the Cyprus dispute: i.e., that objects were carried by refugees not during the flight but following the return. This specific reference has been deleted in the newer version.
- The second meaning of the term, as was seen in the conclusion, addressed the different generational meanings. That is, that according to the ‘politics of memory’, she should have reclaimed the object. This reference has been deleted also in order to avoid confusion.

Reviewer 2 Report
Comments and Suggestions for Authors
Author Response
Response to reviewer:
Firstly, I would like to thank the reviewer for their comments. All are well-received and I believe the new version of the paper, which incorporates these points, is much stronger. See below:
- I have re-contextualised the entire discussion and analysis of the paper. I agree that too many points and discussions were running parallel in the paper, without making a clear contribution to the literature. The new version of the paper focuses on: a) the conflicting interpretations of the story/narration about the mirror piece between my mother and me and how these interpretations were influenced by the politics of memory on the island prevalent when I was growing up, and b) the way the photographic act is connected to refugees’ understanding of the island’s future. These two discussions have their own sub-sections (‘the return’ for a; ‘the photograph’ for b). It must also be noted that the sub-section ‘the return’ provides more context for the politics of memory and describes its main parameters.
- As far as the second issue identified by the reviewer, I cannot claim that the experiences described within the paper are representative of what occurs within each and every family of Greek-Cypriot refugees. Indeed, a study into the history of memory within refugee families will almost certainly yield a multiplicity of different results. I do believe however the new version of the paper makes an argument regarding the politics of memory on the island and how that has affected the intergenerational interpretation of history, whether it be the national history presented in books and public debates or the familial history passed down within families.
- Clarity issues:
- Lines 29-30: I have deleted the reference to “customary memory practices” as indeed it was unclear. The original version of the paper indicated as ‘customary’ practices the fact that material objects are typically carried by refugees during the flight rather than following the return.
- Line 48: I have changed the wording here to make the argument clearer.
- Redundancy issues:
- Lines 104 and after: I understand the reviewer’s comment and I have adjusted this part to reflect the point.
- Extrapolation issues:
- I have deleted the reference to ganimet in the new version as it does not fit in well with the new analysis. This was a point raised by all reviewers.

Reviewer 3 Report
Comments and Suggestions for Authors
This is a fluidly written article that takes a small material item as the starting point for unravelling experiences of long-term displacement in Cyprus. The author focuses on an item that the author’s mother found during a visit to her natal village but which she was unable to recover. Instead, she took a photograph of it, and the author analyses this difference between the photograph and the thing itself in terms of making sense of their displacement and inability fully to return. I believe the article can be published with some revisions, mostly for clarity:
1) The article presents an analysis of one material item’s circulation and documentation, using an article by Bryant that discusses temporality and material items in conflict. What, though, is the larger point that the author wants to make? Bryant’s article discusses Turkish Cypriot appropriation of items and how their meaning changed with the opening of the checkpoints. This author is discussing the Greek Cypriots who lost those items and attempts (or not) to regain or document them. What larger contribution does this discussion make to the literature on materiality, conflict, and time?
2) Furthermore, in the analysis itself, we are not given an explanation of the meaning of the mirror to the family, apart from it having been a gift to the father and being recognizable. Perhaps the mother’s reluctance to push for its return was because it was not as meaningful as the author suggests? Or is the author saying that recovering such an obviously identifiable object would have given the family a connection to the home?
3) There should be further explanation of the author’s choice to conduct research with their family. Although there is reflection on the ethical and methodological problems and advantages presented by this choice, why was it made in the first place?
4) On page 2, the author says, “The ability to temporarily return following the opening of the checkpoints has made the practices associated with material items caught in the dispute rather uncommon.” Two sentences later, however, they remark that “many Greek Cypriot refugees rediscovered lost family objects and heirlooms and often sought to reclaim them.” These sentences are contradictory, and the author needs to decide which is the case.
5) It is not clear what the distinction between remainders and reminders does in this paper. Bryant uses ‘remainders’ to talk about how, for Turkish Cypriots, material items like tables and chairs do not immediately evoke their previous owners and thereby become uncanny reminders when the owners point out that they were theirs. Looked at from the perspective of the Greek Cypriot owners, it’s not clear how such items are remainders. Indeed, even from the perspective of the Turkish Cypriot woman who had appropriated the mirror piece, it seems like a personal enough item that it would be what Bryant calls a ‘remain’, something with a direct attachment to its owner. Although the examples Bryant uses are items like photographs and clothing, the mirror with the ‘Kalimera’ written on it seems quite personal, something that literally speaks. Why call it a remainder here? And what does it become a reminder of? Only of the life they once had? Or perhaps it’s a reminder of other histories that are also entangled with the mother’s reluctance to press for its return?
6) The discussion of ganimet on p. 7 needs to be more nuanced. Ganimet means spoils of war, but the idea that it represents a history come full circle is an interpretation of how TCs justify taking that property. It is not what the word literally means.
7) On p. 8, the author says that they regret not seeking more information. Given that this is the author’s mother, why did the author not do that? Is the mother no longer alive?
Author Response
Response to reviewer
Firstly, I would like to thank the reviewer for their comments. All are well-received and I believe the new version of the paper, which incorporates these points, is much stronger. See below:
- I have re-contextualised the entire discussion and analysis of the paper. I agree that too many points and discussions were running parallel in the paper, without making a clear contribution to the literature. The new version of the paper focuses on: a) the conflicting interpretations of the story/narration about the mirror piece between my mother and me and how these interpretations were influenced by the politics of memory on the island prevalent when I was growing up, and b) the way the photographic act is connected to refugees’ understanding of the island’s future. These two discussions have their own sub-sections (‘the return’ for a; ‘the photograph’ for b). It must also be noted that the sub-section ‘the return’ provides more context for the politics of memory and describes its main parameters.
- I have added further explanation to address this point. As pointed out in the new version, the mirror piece was the only material object connected with the family’s history discovered during the return journey. Additionally, the overall experience of returning to the village was quite traumatic for family members as they were not allowed to enter their former house by its current occupants. Both of these points suggest that the rediscovery of the mirror piece acquired added meaning for family members.
- I have added further detail to address this point in the methodological section.
- I have addressed this point by deleting the first quote. In the initial version the quote was meant to denote that memorial practices following 2003 were uncommon in comparison to other conflict situations but I understand why this might have caused confusion.
- I understand this point and, as indicated above in point 1, the whole discussion has been re-contextualised to focus on two points. The discussion about ‘remainders’ is only briefly discussed to refer to the meanings of the object for the Turkish Cypriot woman, as elaborated upon through a quote from the interview with another one of my aunts. See line 402 and after.
- Same as above. This was a point raised by all reviewers, that the discussion on ganimet was confusing and unclear.
- I understand why this point might have been unclear and I have deleted it in the version. What I regret is failing to ask additional questions during that specific interview. Later during the research project, I managed to gain more information about what occurred during the return journey and the rediscovery of the mirror piece from other family members (which I also share in the new version of the paper). I do believe however that I should have pushed to get more out of my mother during that specific interview as it would have revealed more about her own meanings regarding the rediscovery.

Round 2
Reviewer 1 Report
Comments and Suggestions for Authors
The author revised the manuscript sufficiently. I applaud the improvements. In my opinion, it is ready for publication.
Reviewer 3 Report
Comments and Suggestions for Authors
I previously reviewed this article, and the author has made extensive revisions in response to all reviews. I am satisfied with the author's revisions and recommend publication.